# Calf Birth Weight Predicted Remotely Using Automated in-Paddock Weighing Technology

**DOI:** 10.3390/ani11051254

**Published:** 2021-04-27

**Authors:** Anita Z. Chang, José A. Imaz, Luciano A. González

**Affiliations:** 1School of Life and Environmental Sciences, Faculty of Science, The University of Sydney, Sydney, NSW 2570, Australia; a.chang@cqu.edu.au (A.Z.C.); luciano.gonzalez@sydney.edu.au (L.A.G.); 2Institute for Future Farming Systems, School of Health, Medical, and Applied Sciences, Central Queensland University, Rockhampton North, QLD 4702, Australia; 3Sydney Institute of Agriculture, The University of Sydney, Sydney, NSW 2570, Australia

**Keywords:** remote, liveweight, predictions, calf birth weight

## Abstract

**Simple Summary:**

The use of ‘in paddock’ walk-over-weighing scales (WOW) enables cattle liveweight (LW) data collection, remotely and individually, at a high frequency (e.g., daily). Liveweight data obtained can be used to calculate the liveweight change experienced by a dam at calving (ΔLWC), which is linked with calf birth weight (CBW). This study utilised WOW technology to investigate the degree of association between CBW, LW before and after calving, ΔLWC, and cow non-foetal weight loss at calving (NFW) (ΔLWC–CBW = NFW, e.g., membranes, fluids). Such outcomes could contribute to predicting CBW by assessing ΔLWC without the need of weighing the calf manually, which can be a labour and time-consuming operation in the extensive conditions of beef production. There was no correlation between CBW and the LW of the dam before and after calving; however, positive associations between CBW, ΔLWC, and NFW were found. Particularly, 56% of the variation in ΔLWC was attributed to CBW. These findings suggest that the remote monitoring of ΔLWC has potential to be used for CBW predictions, regardless of the LW of the dam around calving time.

**Abstract:**

The present study aimed to develop predictive models of calf birth weight (CBW) from liveweight (LW) data collected remotely and individually using an automated in-paddock walk-over-weighing scale (WOW). Twenty-eight multiparous Charolais cows were mated with two Brahman bulls. The WOW was installed at the only watering point to capture LW over five months. Calf birth date and weight were manually recorded, and the liveweight change experienced by a dam at calving (ΔLWC) was calculated as pre-LW minus post-LW calving. Cow non-foetal weight loss at calving (NFW) was calculated as ΔLWC minus CBW. Pearson’s correlational analysis and simple linear regressions were used to identify associations between all variables measured. No correlations were found between ΔLWC and pre-LW (*p* = 0.52), or post-LW (*p* = 0.14). However, positive associations were observed between ΔLWC and CBW (*p* < 0.001, R2 = 0.56) and NFW (*p* < 0.001, R2 = 0.90). Thus, the results suggest that 56% of the variation in ΔLWC is attributed to the calf weight, and consequently could be used as an indicator of CBW. Remote, in-paddock weighing systems have the potential to provide timely and accurate LW data of breeding cows to improve calving management and productivity.

## 1. Introduction

Several constraints could preclude beef producers from regularly interacting with their cattle, including limiting weather conditions, the need for increasing labour, and the extensiveness which characterizes large beef properties where stocking rates can be lower than one animal every 150 ha [1,2]. Particularly, such constraints could prevent recording key information to improve reproductive efficiency and productivity of breeding cows, such as calf birth weight (CBW). In this regard, CBW data is logistically difficult to record in the extensive conditions of beef production compared to more intensive systems, demanding continuous monitoring of breeding cows during the calving season to detect calf births, and manually weigh and tag them with electronic identification (EID) [2,3]. Fortunately, advances in novel ‘in paddock’ technologies can be used to record cattle liveweight (LW), continuously, automatically, and remotely on individual animals [2,4,5,6,7,8]. Studies using remote walk-over weighing (WOW) did explore growth trajectories of beef weaners and steers and calving date was estimated in breeding cows [2,9]. In this line, as the LW change experienced by the cow at calving time (ΔLWC) is associated with the LW of its calf, remote weighing could be potentially used to estimate CBW by assessing ΔLWC. This remains important in extensive conditions because measuring the LW of the calf using WOW before marking would not be feasible, as calf EIDs are, in general, not distributed at branding.

Calf birth weight is one of the key factors defining calf morbidity and survival at calving time, but also calf growth and health in later life [10,11,12,13]. For instance, the incidence of dystocia was linked with high CBW and could increase calf mortality at calving and up to 30 days following birth, or compromise long-term health [10,14,15]. Therefore, CBW predictions could be used to maximise cow and calf welfare at birth, alongside avoiding losses associated with low and high CBW. However, CBW is a complex trait influenced by environmental and genetic components [16]. In this regard, ‘in utero’ maternal nutrition is crucial in foetal and placental development throughout gestation, even during the earliest stages of pregnancy, when the nutritional requirements from the foetus are minimal, but placental development, cellular differentiation, vascularization, and foetal organogenesis occur [17,18,19]. Likewise, maternal nutrition affects the weight of the embryonic components, such as the membranes and placentomes [20,21]. Variations in membrane weight exist due to protein content in diets, with high protein resulting in heavier membranes [21]. As such, the weight of the embryonic components can act as a proxy for maternal nutrition, with heavier non-foetal weight loss (NFW) at calving being indicative of a better diet in utero.

As far as we know, there are no investigations on the associations between CBW, NFW, and ΔLWC of breeding cows using LW data collected remotely. These outcomes could be utilised to develop the required predictive models for use in WOW systems where the immediate LW of the calf cannot be readily measured. Thus, the current study aims to assess if a correlation occurs between ∆LWC and CBW, NFW, cow body condition score at calving (BCS), and cow LW pre- and post-calving (pre-LW and post-LW, respectively). It is hypothesised that liveweight change at calving will have a significant positive correlation with CBW, NFW, BCS, and pre-LW, but a negative correlation with post-LW.

## 2. Materials and Methods

### 2.1. Cattle and Management

The study was conducted from June 2017 to October 2017 at John Bruce Pye Farm (The University of Sydney, Sydney, NSW, Australia), where the predominant pasture species grazed included kangaroo grass (*Themeda australis*) and weeping grass (*Microlaena stipoides*). Twenty-eight multiparous Charolais cows (initial pre-calving LW = 715.2 ± 9.45 kg) were joined with two Brahman bulls. The 9.8 ha paddock was checked in the morning and afternoon of each day to locate any calves born. Calves were ear-tagged with a visual ear tag and an electronic identification ear tag (DNA Ear tag, Allflex Australia Pty, Capalaba, QLD, Australia) and weighed using the Gallagher Weigh Scale and Data Recorder W810 v2 in conjunction with a calf birth weight cradle (Ramage Engineering, South Guyra, NSW, Australia). The cow identification, birth date, sex, and cow BCS at calving were recorded.

### 2.2. Walk-Over-Weighing Station

A yard of approximately 30 m × 30 m was constructed around the sole watering point within the paddock to maintain a high frequency of cattle visitation to the WOW system, as proposed by González, Bishop-Hurley, Henry, and Charmley [7] (Figure 1). A WOW cattle platform (XR300 model, Tru-Test, Eight Mile Plains, QLD, Australia) was installed at the entry of the confined watering point to record LW as the animals accessed water but to ensure LW data was unaffected by water consumption [7]. The weighing system consisted of an electronic scale, an EID panel reader to identify animals utilizing the system, data storage and cellular communication, and entry and exit spear gates to allow one-directional movement of animals through the WOW system. Raw LW data gathered was remotely accessed and downloaded weekly. Cows were familiar with the weighing system, as the system had been in place for three years before the experiment, so behavioural aversions to the system did not affect data collection.

### 2.3. Statistical Analysis

Data recorded by the WOW were filtered for outlying data and then smoothed using penalised b-splines using the methods described by [7]. The resulting cow LW data were averaged by date for each animal if more than one measurement per day and animal existed. Then, this data was used to calculate ΔLWC using the difference between the average LW of the week before minus the average LW of the week following calving. Daily records from the week before and after calving were considered to avoid LW variability and the lack of attendance of some animals to the water point around calving time [22]. Non-foetal weight (NFW) was calculated by subtracting the recorded CBW from ΔLWC. The remaining filtered data was analysed using RStudio (RStudio Inc., Boston, MA, USA) and Microsoft Excel (Microsoft Corporation, Redmond, WA, USA). A Pearson’s correlational matrix was used to identify stepwise correlations between all variables measured. Simple linear regressions were used to determine the relationships between ΔLWC and CBW and to obtain the regression equations. Statistical significance was declared at *p* < 0.05.

## 3. Results

Cow liveweight change at calving averaged 80.50 ± 3.97 kg, with a mean of 36.77% being due to NFW and 63.23% to CBW (Table 1). In addition, a broad range of ΔLWC was observed between individual animals (48.1 to 134.0 kg/hd). Foetal weight loss at calving showed the greatest variation across all variables explored in cows (CV = 54.38%), with ΔLWC showing the second largest variation (=26.12%; Table 1). Figure 2 presents data from the remote weighing station of two individual cows experiencing high and low ΔLWC (panels a and b, respectively) as examples to illustrate the filtered LW data and liveweight change throughout the study. In both cases, calving occurred when liveweight change was at its highest peak.

Positive correlations were identified between ΔLWC and NFW (*p* < 0.001; R = 0.95) and CBW (*p* < 0.001, R = 0.76), as well as CBW and NFW (*p* = 0.004; R = 0.52; Table 2). As expected, pre-LW was positively correlated with post-LW (*p* < 0.001; Table 2). Surprisingly, BCS was negatively correlated with CBW (*p* = 0.05; Table 2).

Table 3 presents the regression equations to predict CBW and ΔLWC from each individual independent variable. A positive relationship was identified between ΔLWC and both NFW (*p* < 0.001) and CBW (*p* < 0.001), with the former contributing the largest proportion of variation in ΔLWC (R^2^ = 0.90) and the latter contributing the second largest proportion of variation (R^2^ = 0.56; Table 3). Thus, body weight loss of cows at calving (ΔLWC) increased by 2.12 ± 0.36 kg for every unit of increase in CBW and by 1.25 ± 0.08 kg for every unit of increase in NFW (*p* < 0.001; Table 3). However, ΔLWC was unaffected by pre-LW, post-LW, and BCS (*p* > 0.05; Table 3). Calf birth weight increased by 0.27 ± 0.05 kg for every kilogram increase in cow ΔLWC (*p* < 0.001; Table 3). However, CBW decreased with increasing cow BCS (*p* = 0.05), whereas no relationship was found between CBW and pre-LW or post-LW (*p* > 0.05; Table 3).

## 4. Discussion

We hypothesized in the present study that the ΔLWC of breeding cows, calculated using LW data collected remotely, was positively correlated with CBW, NFW, BCS, and pre-LW but negatively associated with post-LW. The existence of such associations in combination with near-real-time LW data collected around calving would enable us to build models to predict CBW in situations when interacting with cattle is not logistically feasible. Results from the present study support our hypothesis as positive associations were observed between ΔLWC, CBW, and NFW. However, pre-LW and post-LW did not explain CBW variability, which was negatively correlated with BCS. In this regard, CBW was less variable and represented the largest proportion of ΔLWC, whereas NFW largely differed amongst animals. In line with our findings, other studies reported that CBW was less prone to variations than NFW which, in turn, could be affected by the weight of embryonic membranes, variations in the foetal fluid volume, and feed and water intake close to calving [16,20,21]. Additionally, maternal behaviour around calving could affect the number of days with LW measurements (e.g., lack of attendance to the water point) and thus ΔLWC calculations. However, as far as we know, no studies were reported using remotely collected LW data to predict CBW to compare such outcomes with the associations found in the present study.

### 4.1. Components of Cow Liveweight Loss at Calving

Cow liveweight loss at calving can be broadly broken down into CBW and NFW. In the present study, a broad range of ΔLWC was observed between individual animals (CV = 26.12%). As expected, NFW and CBW were positively correlated with ΔLWC (*p* < 0.001). Non-foetal weight contributed to 36.8% of ΔLWC and explained the greatest proportion of its variability (R2 = 90%). Conversely, CBW represented the largest proportion of the ΔLWC (63.23%) but showed lower variability amongst animals compared to NFW. The variation in NFW could be attributable to different factors, which included the weight of embryonic membranes, external factors such as variations in the foetal fluid volume, and internal factors such as the uterine capacity of the dam [16]. Nevertheless, based on the methodology employed in this study, it was not possible to determine the exact contribution of each NFW factor to ΔLWC. On the other hand, CBW was less variable than NFW. This could be in part because CBWs that are significantly higher or lower than average are often not conducive to survival [16]. Underweight calves are generally less vigorous, with reduced cold tolerance and increased disease susceptibility, while those that are overweight are at risk of experiencing dystocia due to foeto-pelvic disproportion and are more likely to experience increased perinatal mortality [16,23,24]. Consequently, excessively heavy or light calves do not reach reproductive age and are unable to pass on the trait for birth weight. Therefore, CBW seems to be a more conservative measure amongst animals than NFW as the latter may be more variable. However, further controlled studies should be conducted to determine the contribution of potential factors influencing NFW which would aid to develop more precise prediction models of CBW.

The positive relationship between NFW and CBW supports our hypothesis that a positive correlation would be identified between the two variables. It could be suggested that the extraembryonic components making up NFW have a significant contribution to foetal development and consequently to CBW, supported by previous findings by Echternkamp [25] and Sullivan, Micke, Magalhaes, Phillips, and Perry [20]. These studies reported on the positive correlation between CBW and placental weight in beef cattle and attributed its effect to the role of these components in embryo development. However, the exact contribution of each component of NFW to CBW remains unclear. This knowledge would enable the manipulation of CBW, for example, through factors influencing NFW.

### 4.2. Use of Models in Conjunction with Walk-Over-Weighing Systems

The models developed in the present study identifying the relationship between ΔLWC and CBW would be best utilized in extensive production systems, where high calf mortality is commonplace and significant challenges exist that impede calving data collection and regular interaction with livestock [1,2]. Accurate and complete calving data is essential for improving reproductive management and maximizing genetic gain which could lead to enhancing productivity in beef production systems. Consequently, WOW systems have been introduced in many of these properties to alleviate some of the monitoring and management issues. However, the WOW system requires an RFID tag to be administered to match LW data with a certain animal. Due to the pre-existing challenges that prevent regular interaction with livestock, it is thus not always possible to allocate the calf an RFID ear tag soon after birth. Therefore, there is still difficulty in measuring early life data directly from the calf. However, the present study results can be used to elucidate CBW from ΔLWC, which cannot be predicted with great accuracy and requires extensive amounts of labour, data, and analysis [26,27]. Thus, by utilizing WOW technology in conjunction with prediction models, such as those developed in the present study, the proportion of weight lost at calving can be automatically estimated and attributed to NFW and CBW, which in turn can be used to enhance management decisions such as reducing calf mortality and increasing genetic gain under grazing conditions.

### 4.3. Cow Traits

It was hypothesised that increased BCS would result in greater NFW due to heavier extraembryonic components; however, the opposite relationship was observed in this study (*p* < 0.05). Extensive scientific literature exists that describes the effect of varying planes of maternal nutrition on the weight of the membranes, with more nutritious diets resulting in heavier membranes and cotyledons [28,29]. Likewise, it was expected that the improved maternal nutrition resulting in greater embryonic membrane weight would, in turn, increase the CBW [20,21]; however, a negative relationship was observed between CBW and BCS (*p* = 0.05). Therefore, it could be suggested that the majority of NFW is not attributable to embryonic membranes, but the contribution is unknown and further studies must be conducted. Additionally, the BCS measured at calving may not represent the BCS and nutritional conditions experienced by the cow throughout gestation. To more accurately determine the influence of BCS on NFW and CBW, more frequent monitoring of BCS throughout pregnancy is necessary. Further studies could be conducted to impose nutritional treatments to more closely control BCS.

Feed and water intake on days before and after calving proved to be variable not only over time but also between individual animals [16,22]. These parameters tend to decrease before calving, before increasing again following parturition [30,31]. Potential increases in food and water consumption after calving, but before using the WOW system, could mask or skew the observed ΔLWC. Additionally, as described by Aldridge, Lee, Taylor, Popplewell, Job, and Pitchford [2], if a calf suckles before the cow uses the WOW, this could also affect ΔLWC recorded by the WOW system. In the same line, cows’ attendance to the water point, and the possibility of weighing them automatically, could be reduced due to the behaviour of the dam before calving, while preparing to calve, or after calving because of nursing activities [32,33]. The exact contribution of each of these components to NFW is unknown, and therefore further research should aim to investigate their influence on NFW to elucidate reasons for such variations in ΔLWC.

The liveweight of the cow before calving, as well as the BCS of the cow at calving, were not found to have a significant impact on ΔLWC in the present study. This study, however, utilized cattle of the same breed and a genetic or breed effect may exist in other experimental conditions. Nevertheless, based on our results, it is unknown what cow-based factors contribute to the significant variation in ΔLWC and consequently, we are unable to predict ΔLWC using pre-LW or BCS, and suggest that CBW cannot be manipulated using pre-LW. The present study also suggests that the complexities of ΔLWC and its relationship with both cow- and calf-based traits have not been revealed, but they suggest that LW around the time of calving, i.e., pre- and post-LW, may have a minimal or non-existent impact on ΔLWC. This indicates that remote monitoring of ΔLWC to predict calf birth weight does not need to account for BCS or the nutritional status of the cow. Nonetheless, additional factors which may have a greater impact on ΔLWC need to be investigated.

## 5. Conclusions

Remote, in-paddock weighing systems demonstrated potential to provide timely and accurate LW data of breeding cows to develop prediction models of calf birth weight. These models could be used in conjunction with weighing systems to improve calving management and productivity.

## Figures and Tables

**Figure 1 animals-11-01254-f001:**
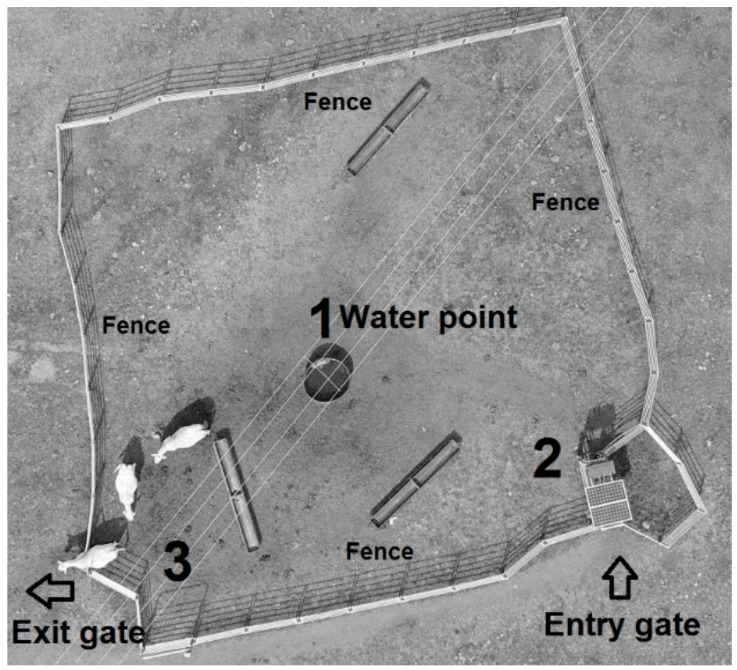
Aerial photo (drone) showing the setup of the walk-over-weighing system. Numbers indicate (**1**) water point; (**2**) walk-over-weighing scale deployed at the entry gate; and (**3**) exit gate. Both entry and exit gates enabled cattle to walk through the unidirectional spear gates to access the water trough.

**Figure 2 animals-11-01254-f002:**
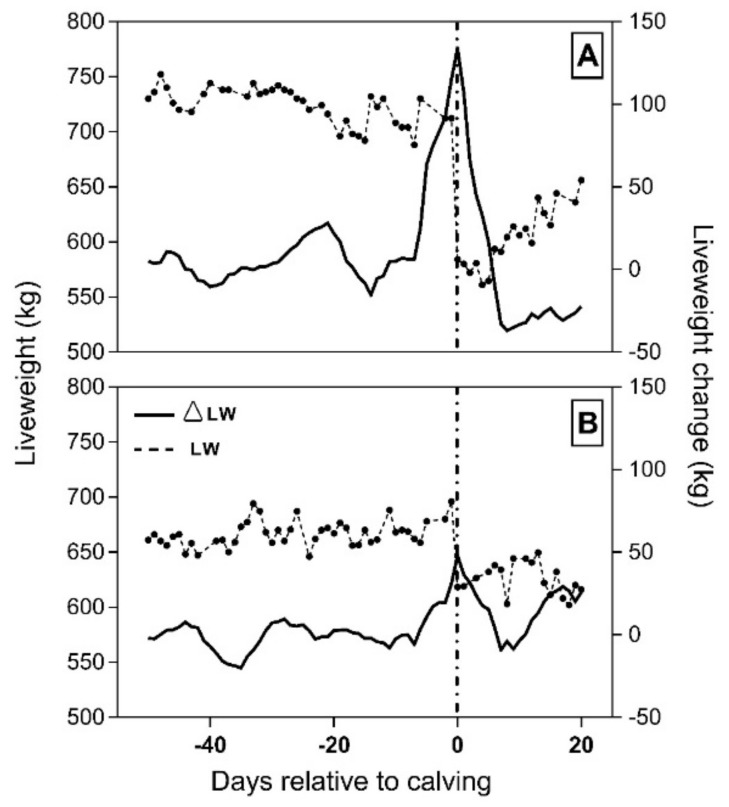
Liveweight (connected dots) and liveweight change over 7 days (solid line) for a cow experiencing high weight loss (**A**) and low weight loss (**B**) at calving. A single dot represents the daily averaged LW for both cows. Calving (dashed vertical reference line) occurred for (**A**) on 29 August 2017 when ∆LWC peaked at 112.90 kg and for (**B**) on 20 August 2017 when ∆LWC peaked at 60.43 kg.

**Table 1 animals-11-01254-t001:** Descriptive statistics (mean, minimum, maximum, s.d. -standard deviation-, s.e. -standard error-, c.v. -coefficient of variation-) of cow liveweight change at calving (ΔLWC), cow non-foetal weight loss at calving (NFW), cow pre-calving liveweight (pre-LW), cow post-calving liveweight (post-LW), and calf birth weight (CBW) (*n* = 28).

Statistic	∆LWC (kg)	CBW (kg)	NFW (kg)	Pre-LW (kg)	Post-LW (kg)	BCS
Mean	80.50	50.90	29.60	719.20	639.10	2.5
Minimum	48.10	38.80	4.10	630.60	534.30	2.0
Maximum	134.00	66.00	68.50	829.70	742.00	3.5
s.d.	21.03	7.53	16.08	49.98	52.28	0.35
s.e.	3.97	1.42	3.04	9.45	9.88	0.07
c.v. (%)	26.12	14.79	54.38	6.99	8.18	13.90

**Table 2 animals-11-01254-t002:** Pearson’s correlation matrix between cow liveweight change at calving (∆LWC), calf birth weight (CBW), non-foetal weight (NFW), cow pre-calving liveweight (pre-LW), and cow post-calving liveweight (post-LW). Values above the diagonal are correlation coefficients, with *p*-values indicated below the diagonal.

Variable	∆LWC	CBW	NFW	Pre-LW	Post-LW	BCS
∆LWC	1	0.76	0.95	0.12	−0.28	−0.32
CBW	<0.001	1	0.52	0.28	−0.12	−0.37
NFW	<0.001	0.004	1	0.02	−0.32	−0.25
Pre-LW	0.56	0.14	0.93	1	0.84	−0.01
Post-LW	0.14	0.56	0.10	<0.001	1	0.07
BCS	0.10	0.05	0.21	0.98	0.73	1

**Table 3 animals-11-01254-t003:** Prediction equations for cow liveweight change at calving (ΔLWC) and calf birth weight (CBW) from the non-foetal weight (NFW), cow pre-calving liveweight (pre-LW), and cow post-calving liveweight (post-LW) in the 28 cow–calf pairs, alongside the regression coefficient (β), intercept (α), R^2^, standard error (s.e.) and *p*-value for the intercept regression coefficient and model.

Items	Intercept		Regression Coefficient		Model	
	α ± s.e.	*p*-Value	β ± s.e.	*p*-Value	R^2^	*p*-Value
∆LWC						
CBW	−27.38 ± 18.35	0.15	2.12 ± 0.36	<0.001	0.56	<0.001
NFW	43.66 ± 2.63	<0.001	1.25 ± 0.08	<0.001	0.90	<0.001
Pre-LW	45.94 ± 58.75	0.44	0.05 ± 0.082	0.56	−0.02	0.56
Post-LW	153.30 ± 48.50	0.004	−0.11 ± 0.08	0.14	0.04	0.14
BCS	130.01 ± 28.85	<0.001	−19.46 ± 11.23	0.10	0.07	0.10
CBW						
∆LWC	29.04 ± 3.81	<0.001	0.27 ± 0.05	<0.001	0.56	<0.001
NFW	43.66 ± 2.63	<0.001	0.25 ± 0.08	0.004	0.25	0.004
Pre-LW	29.94 ± 20.32	0.324	0.04 ± 0.03	0.14	0.04	0.14
Post-LW	61.62 ± 17.99	0.002	−0.02 ± 0.03	0.56	−0.02	0.56
BCS	71.56 ± 10.12	<0.001	−8.12 ± 3.94	0.05	0.11	0.05

## Data Availability

Data utilised in the present study are available from the corresponding author on reasonable request.

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
