# Peer review of "Calf Birth Weight Predicted Remotely Using Automated in-Paddock Weighing Technology"

_animals, 2021, doi:10.3390/ani11051254_

Round 1
Reviewer 1 Report
This study utilized WOW technology to make correlations between known cow weights to known calf birth weights in order to predict calf birth weights from the dataset. Authors used Microsoft Excel to compute statistical data. This is an interesting use of WOW technology, but very important from a producer standpoint. If more cattle subjects had been used, perhaps the correlations would have appeared as expected. The only thing lacking is a needed discussion on animal weight variability that possibly effects the predicted outcome from what was reported.
Some suggested edits for increased readability:
Line 16: Please define first use of NFW, to be the same as Line 30
Line 34: Should say calf weight instead of weight calf.
Line 44: “Avoid” is used incorrectly – suggest another word, like preclude.
Line 56 – 57: Please rewrite, it is a confusing statement.
Line 61: Omit the word manipulating.
Line 77: Do you mean to say the immediate live weight of the calf? Please edit as such.
Line 122: Should it say “calving” instead of “birth”, because the cow weight is being considered in this case, not the calf weight?
Line 152: Should say “vertical” instead of “horizontal”.
Line 158: Correct “surprisingly”.
Line 194: Use “least” instead of “latest”.
Line 256-257: reducing calf mortality increasing and genetic gain under grazing conditions. Do you mean “and increasing” instead of “increasing and”?
Line 279: Replace “using” with “uses”.
Author Response
See attached file. Thanks for your review,

Reviewer 2 Report
The paper was well written and clear. The background and rationale was well communicated and it identified a potential issue that I had not been aware of. The statistical analysis was thorough.
I would recommend the paper is published
Author Response
See attached, thanks for your review

Reviewer 3 Report
animals-1152835-peer-review-v1
Calf birth weight predicted remotely using automated in-paddock weighing technology
This article represents a contribution to scientific knowledge about the use of a remote system for weighing cows and, in this way, to develop predictive models for calf weight and other traits at calving. Thus, it is expected that there will be timely and accurate calf management.
The article is well organized and written clearly. Despite this, some improvements can be made to make the text clearer. The text addresses the subject with scientific correctness. The figures and tables are relevant and are well presented; the methods are clearly described, which will allow a perfect understanding by other researchers. The results are well discussed with the existing knowledge on the subject. The results support the conclusions.
Some detailed comments below:
L16 spell out NFW
L158 Surpriingly, “please change with” Surprisingly
L189 LW, but “please change with” LW but
L193 and NFW, however, “please change with” and NFW. However,
L194 explain variability in CBW, “please change with” explain CBW variability,
L195 of ΔLWC whereas NFW largely “please change with” of ΔLWC, whereas NFW largely
L212 factors which included “please change with” factors, which included
L229-230 to fetal development, and consequently to CBW, which is supported by previous “please change with” to fetal development and consequently to CBW, supported by previous
L235 enable to manipulate CBW “please change with” enable to manipulation of CBW
L246-252 However, the results from the present study can be used to elucidate CBW from ΔLWC, which at present time cannot be predicted with great accuracy and requires extensive amounts of labour, data, and analysis [26,27]. “please change with” However, the present study results can be used to elucidate CBW from ΔLWC, which cannot be predicted with great accuracy and requires extensive amounts of labour, data, and analysis [26,27].
L261 components, however, the opposite “please change with” components; however, the opposite
L283 NFW is unknown and therefore further “please change with” NFW is unknown, and therefore further
L292-295 Results from the present study also suggest that complexities of ΔLWC and its relationship with both cow- and calf-based traits have not been revealed but they suggest that LW around the time of calving, i.e. pre- and post-LW, may have a minimal or non-existent impact on ΔLWC. “please change with” The present study also suggests that complexities of ΔLWC and its relationship with both cow- and calf-based traits have not been revealed but suggest that LW is around the time of calving, i.e. pre- and post-LW, may have a minimal or non-existent impact on ΔLWC.
Author Response
See attached file, thanks for your review
